# EMERGENT TRANSLATION
# IN MULTI-AGENT COMMUNICATION

**Jason Lee**[*]
New York University
jason@cs.nyu.edu

**Kyunghyun Cho**
New York University
Facebook AI Research
kyunghyun.cho@nyu.edu

**Jason Weston**
Facebook AI Research
jase@fb.com

**Douwe Kiela**
Facebook AI Research
dkiela@fb.com

## ABSTRACT

While most machine translation systems to date are trained on large parallel corpora, humans learn language in a different way: by being grounded in an environment and interacting with other humans. In this work, we propose a communication game where two agents, native speakers of their own respective languages, jointly learn to solve a visual referential task. We find that the ability to understand and translate a foreign language emerges as a means to achieve shared goals. The emergent translation is interactive and multimodal, and crucially does not require parallel corpora, but only monolingual, independent text and corresponding images. Our proposed translation model achieves this by grounding the source and target languages into a shared visual modality, and outperforms several baselines on both word-level and sentence-level translation tasks. Furthermore, we show that agents in a multilingual community learn to translate better and faster than in a bilingual communication setting.

## 1 INTRODUCTION

Building intelligent machines that can converse with humans is a longstanding challenge in artificial intelligence. Remarkable successes have been achieved in natural language processing (NLP) via the use of supervised learning approaches on large-scale datasets (Bahdanau et al., 2015; Wu et al., 2016; Gehring et al., 2017; Sennrich et al., 2017). Machine translation is no exception: most translation systems are trained to derive statistical patterns from huge parallel corpora. Parallel corpora, however, are expensive and difficult to obtain for many language pairs. This is especially the case for low resource languages, where parallel texts are often small or nonexistent. We address these issues by designing a multi-agent communication task, where agents interact with each other in their own native languages and try to work out what the other agent meant to communicate. We find that the ability to translate foreign languages emerges as a means to achieve a common goal.

Aside from the benefit of not requiring parallel data, we argue that our approach to learning to translate is also more natural than learning from large corpora. Humans learn languages by interacting with other humans and referring to their shared environment, i.e., by being *grounded* in physical reality. More abstract knowledge is built on top of this concrete foundation. It is natural to use vision as an intermediary: when communicating with someone who does not speak our language, we often directly refer to our surroundings. Even linguistically distant languages will, by physical and cognitive necessity, still refer to scenes and objects in the same visual space.

We compare our model against a number of baselines, including a nearest neighbor method and a recently proposed model (Nakayama & Nishida, 2017) that maps languages and images to a shared space, but lacks communication. We evaluate performance on both word- and sentence-level translation, and show that our model outperforms the baselines in both settings. Additionally, we show

---

[*]Work done while the author was interning at Facebook AI Research.

that multilingual *communities* of agents, comprised of native speakers of different languages, learn faster and ultimately become better translators.

## 2 PRIOR WORK

Recent work has used neural networks and reinforcement learning in multi-agent settings to solve a variety of tasks with communication, including simple coordination (Sukhbaatar et al., 2016), logic riddles (Foerster et al., 2016), complex coordination with verbal and physical interaction (Lowe et al., 2017), cooperative dialogue (Das et al., 2017) and negotiation (Lewis et al., 2017).

At the same time, there has been a surge of interest in communication protocols or languages that emerge from multi-agent communication in solving these various tasks. Lazaridou et al. (2017) first showed that simple neural network agents can learn to coordinate in an image referential game with single-symbol bandwidth. This work has been extended to induce communication protocols that are more similar to human language, allowing multi-turn communication (Jorge et al., 2016), adaptive communication bandwidth (Havrylov & Titov, 2017) and multi-turn communication with a variable-length conversation (Evtimova et al., 2017), and simple compositionality (Kottur et al., 2017; Mordatch & Abbeel, 2017). Meanwhile, Andreas et al. (2017) proposed a model to interpret continuous message vectors by "translating" them.

Our work is related to a long line of work on learning multimodal representations. Several approaches proposed to learn a joint space for images and text using Canonical Correlation Analysis (CCA) or its variants (Hodosh et al., 2013; Andrew et al., 2013; Chandar et al., 2016). Other works minimize pairwise ranking loss to learn multimodal embeddings (Socher et al., 2014; Kiros et al., 2014; Ma et al., 2015; Vendrov et al., 2015; Kiela et al., 2017). Most recently, others extended this work to learn joint representations between images and multiple languages (Gella et al., 2017; Calixto et al., 2017b; Rajendran et al., 2016).

In machine translation, our work is related to image-guided (Calixto et al., 2017a; Elliott & Kádár, 2017; Caglayan et al., 2016) and pivot-based (Firat et al., 2016; Hitschler et al., 2016) approaches. It is also related to previous work on multiagent translation for low-resource language pairs (without grounding) (He et al., 2016a). At word-level, there has been work on translation via a visual intermediate (Bergsma & Van Durme, 2011), including with convolutional neural network features (Kiela et al., 2015; Joulin et al., 2016).

It was recently shown that zero-resource translation is possible by separately learning an image encoder and a language decoder (Nakayama & Nishida, 2017). The main difference to our work is that their models do not perform communication.

## 3 TASK AND MODELS

### 3.1 COMMUNICATION TASK

We let two agents communicate with each other in their own respective languages to solve a visual referential task. One agent sees an image and describes it in its native language to the other agent. The other agent is given several images, one of which is the same image shown to the first agent, and has to choose the correct image using the description. The game is played in both directions simultaneously, and the agents are jointly trained to solve this task. We only allow agents to send a sequence of discrete symbols to each other, and never a continuous vector.

Our task is similar to Lazaridou et al. (2017), but with the following differences: communication (1) is bidirectional and (2) of variable length; (3) the speaker is trained on both the listener's feedback and ground-truth annotations; and (4) the speaker only observes the target image and no distractors.

Let $P_A$ and $P_B$ be our agents, who speak the languages $L_A$ and $L_B$ respectively. We have two disjoint sets of image-annotation pairs: $(I_A, M_A)$ in language $L_A$ and $(I_B, M_B)$ in language $L_B$.

**Task in language $L_A$** : $P_A$ is the speaker and $P_B$ is the listener.

1. A target image and annotation $(i, m) \in \{I_A, M_A\}$ is drawn from the training set in $L_A$.

2. Given $i$, the speaker ($P_A$) produces a sequence of symbols $\hat{m}$ in language $L_A$ to describe the image and sends it to the listener. The speaker's goal is to produce a message that is *both* an accurate prediction of the ground-truth annotation $m$, and helps the listener ($P_B$) identify the target image.

3. $K - 1$ distracting images are drawn from $I_A$ at random. The target image $i$ is added to this set and all $K$ images are shuffled.

4. Given the message $\hat{m}$ and the $K$ images, the listener's goal is to identify the target image.

**Task with language** $L_B$ : The agents exchange the roles and play similarly.

We explore two different settings: (1) a *word-level* task where the agents communicate with a single word, and (2) a *sentence-level* task where agents can transmit a sequence of symbols.

## 3.2 MODEL ARCHITECTURE AND TRAINING

Each agent has an image encoder, a native speaker module and a foreign language encoder. In English-Japanese communication, for instance, the English-speaking agent $P_A$ consists of an image encoder $E_{\text{IMG}}^A$, a native English speaker module $S_{\text{EN}}^A$, and a Japanese encoder $E_{\text{JA}}^A$. Similarly, the Japanese-speaking agent $P_B = (E_{\text{IMG}}^B, S_{\text{JA}}^B, E_{\text{EN}}^B)$.

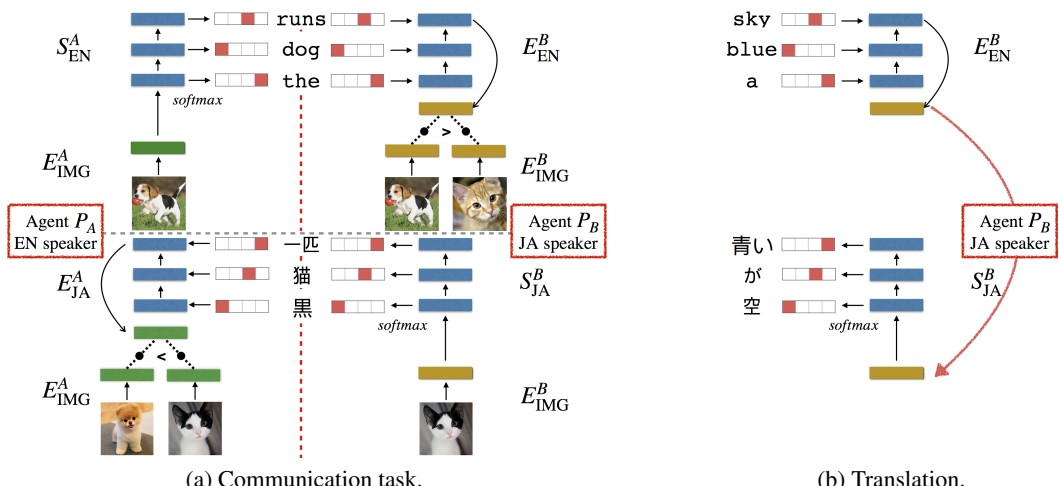

(a) Communication task.          (b) Translation.

Figure 1: Sentence-level communication task and translation between English and Japanese. (a) The red dotted line delimits the agents and the gray dotted line delimits the communication tasks for different languages. Representations residing in the multimodal space of Agent A and B are shown in green and yellow, respectively. (b) An illustration of how the Japanese agent might translate an unseen English sentence to Japanese.

We now illustrate the architecture of our model using the English part of the communication task as an example (upper half of Figure 1a). We first describe the sentence-level model.

**Speaker** ($P_A$) Given an image-annotation pair $(i, m) \in \{I_{\text{EN}}, M_{\text{EN}}\}$ sampled from the English training set, let $i$ be represented as a $D_{\text{img}}$-dimensional vector. $P_A$'s speaker encodes $i$ into a $D_{\text{hid}}$-dimensional vector with a feedforward image encoder: $h_0 = E_{\text{IMG}}^A(i)$.

Our speaker module $S_{\text{EN}}^A$ is a recurrent neural network (RNN) with gated recurrent units (GRU, (Cho et al., 2014)). Our RNN takes the image representation $h_0$ as initial hidden state and updates its state as $h_{t+1} = \text{GRU}(h_t, m_t)$ where $m_t$ is the $t$-th token in $m$. The output layer projects each hidden state $h_t$ over the English vocabulary $\mathbb{V}_{\text{EN}}$, followed by a softmax to predict the next token: $p_t = \text{softmax}(W_o h_t + b_o)$. The speaker's predictions are trained on the ground truth English

annotation $m$ using the cross entropy loss:

$$\mathcal{J}_{\text{spk}}^{\text{EN}} = -\frac{1}{N_{\text{EN}}} \sum_{(i,m)}^{\{I_{\text{EN}}, M_{\text{EN}}\}} \sum_{t=1}^{T_m} \log p(m_t | m_{(<t)}, i),$$

where $N_{\text{EN}}$ is the size of the English training set, and $T_m$ is the length of $m$.

To generate a sequence of tokens, we sample from the categorical distribution $\texttt{Cat}(p_t)$. However, sampling is a non-differentiable computation. To allow our model to be end-to-end differentiable, we use the straight-through Gumbel-softmax estimator (Jang et al., 2017; Maddison et al., 2017) to sample from $\texttt{Cat}(p_t)$ and let the gradient flow, while the speaker sends a sequence of discrete symbols.[1] The message $\hat{m}$ is a sequence of one-hot vectors: $\hat{m} = \{y_t\}_{t=1}^{T_m}$, where $y_t = \texttt{Gumbel\_ST}(p_t)$ is discretized in the forward pass.

**Listener ($P_B$)**   The Japanese-speaking agent $P_B$ encodes the $K$ images into $D_{\text{hid}}$-dimensional multimodal space with its own feedforward image encoder: $\{E_{\text{IMG}}^B(i_k)\}_{k=1}^K$. It also feeds each token from $\hat{m}$ into its English encoder RNN with $D_{\text{hid}}$-dimensional hidden states: $s_{t+1} = \texttt{GRU}(s_t, \hat{m}_t)$. Taking the last hidden state, the representation of $\hat{m}$ is a $D_{\text{hid}}$-dimensional vector: $E_{\text{EN}}^B(\hat{m}) = s_{T_m}$. Note, that encodings of the images and the message have the same dimensionality.

To encourage the listener to align the message representation closest to the target image, it is trained using a cross entropy loss where the logits are given by the reciprocal of the mean squared error (MSE) between the target image and the message representation: $\{1/\big(E_{\text{EN}}^B(\hat{m}) - E_{\text{IMG}}^B(i_k)\big)^2\}_{k=1}^K$.

$$\mathcal{J}_{\text{lsn}}^{\text{EN}} = -\frac{1}{N_{\text{EN}}} \sum_{i \in I_{\text{EN}}} \sum_{\hat{m}} \log\left(\texttt{softmax}\Big(1/\big(E_{\text{EN}}^B(\hat{m}) - E_{\text{IMG}}^B(i)\big)^2\Big)\right) \qquad (1)$$

where the softmax operation is performed over $K$ images. We observed that optimization significantly slows down after the initial stage of learning when training with the standard MSE loss. In order to ensure fast convergence throughout training, we use this modified form of MSE as a loss function whose slope gets steeper as the loss is minimized. See Appendix A for a discussion and a more thorough comparison and analysis.

**Training**   These two agents are jointly trained by minimizing the sum of speaker and listener loss:

$$\mathcal{J} = \sum_{x \in \{\text{EN,JA}\}} (\mathcal{J}_{\text{spk}}^x + \mathcal{J}_{\text{lsn}}^x).$$

Note that the listener is only trained on $\mathcal{J}_{\text{lsn}}$, while the speaker is trained on both $\mathcal{J}_{\text{lsn}}$ and $\mathcal{J}_{\text{spk}}$.

**Word-level model**   The word-level model has a similar architecture to the sentence-level one: instead of an RNN, the speaker module $S_{\text{EN}}^A$ is a feedforward layer that projects $h_0$ over the native vocabulary. We again use a straight-through Gumbel-softmax to sample a one-hot vector. Similarly, the foreign language encoder consists simply of the $D_{\text{hid}}$-dimensional foreign word embeddings.

**General training details**   In both word- and sentence-level experiments, we use 2048-dimensional pre-softmax features from a pre-trained ResNet with 50 layers (He et al., 2016b), instead of raw images. Our models are trained using stochastic gradient descent with the Adam optimizer (Kingma & Ba, 2014). The norm of the gradient is clipped with a threshold of 1 (Pascanu et al., 2013). Gumbel-softmax temperature is tuned on the validation set, but fixed throughout training, not annealed or learned.

## 3.3   How Translation Arises

To translate an English sentence $m_{\text{src}}$ to Japanese, we let the Japanese-speaking agent $P_B$ encode $m_{\text{src}}$ with its English encoder, and decode this representation using its Japanese speaker module:

---

[1] We also trained our models with REINFORCE (Williams, 1992) in our preliminary experiments, but found it to converge much slower than Gumbel-softmax relaxation.

$m_{\text{hyp}} = S_{\text{JA}}^B(E_{\text{EN}}^B(m_{\text{src}}))$ (see Figure 1b). Solving the image referential task requires aligning the foreign (source) sentence representation with the representation of the correct image, which will allow the speaker module to describe the source sentence in its native (target) language, as though it were an image.

## 4 WORD-LEVEL EXPERIMENTS

**Task and dataset**   We train our model on a word-level communication task, where the agent $P_A$ is given an image and needs to find the right word to communicate it so that the agent $P_B$ can pick the right image from a set of distractors. We use the Bergsma500 dataset (Bergsma & Van Durme, 2011), a collection of up to 20 image search results per concept and language, for 500 common concepts across 6 languages: English, Spanish, German, French, Italian and Dutch. We train on 80% of the images, and choose the model with the best communication accuracy on the 20% validation set when reporting translation performance. As the Bergsma500 is an extremely small dataset, we do not have a separate test set to report the communication accuracy on. We only report the translation performance instead. Note that the translation task involves translating 500 words from the vocabulary, therefore the data split of images is not relevant for this task.

**Baselines**   For our baselines, we use a variety of nearest neighbor methods based on similarity metrics in the ConvNet feature space (Kiela et al., 2015). Given a set of 20 ResNet image vectors per concept and language, we can either average them (CNN-Mean) or take the dimension-wise maximum (CNN-Max) to derive a single aggregated image vector. To find the German word for *dog*, for instance, we rank all German words based on cosine similarity between the image vector of *dog* and their image vectors. We then examine precision in retrieving the correct German word, *Hund*. Alternatively, we also consider the similarities between individual image vectors instead of their aggregation: Bergsma & Van Durme (2011) propose taking the average of the maximum similarity scores (CNN-AvgMax) and the maximum of the maximum similarity scores (CNN-MaxMax).

**Experimental settings**   We train with 1 distractor ($K = 2$)[2], learning rate $3\text{e}{-}4$, and minibatch size 128. The embedding and hidden state dimensionalities are set to 400. When the validation accuracies of both the speaker and the listener stop improving, training terminates and we evaluate the performance on the word-level translation task, as described in § 3.3. We consider all 15 language pairs in both directions, reporting results averaged across 30 translation cases.

**Results**   In all 15 language pairs, we observe that translation performance improves with communication performance (Figure 3). In translation, our model outperforms all the nearest neighbor baselines (Table 2). This shows that our agents can learn foreign word representations that are not only effective in solving referential tasks, but also more meaningful than raw image features in translation. It also demonstrates that communication helps to identify and learn correspondences between concepts in different languages.

| Model | P@1 | P@5 | P@20 |
|-------|-----|-----|------|
| CNN-AvgMax | 53.00 | 68.30 | 78.36 |
| CNN-MaxMax | 49.85 | 65.91 | 77.17 |
| CNN-Mean | 51.89 | 66.47 | 77.14 |
| CNN-Max | 33.81 | 50.62 | 65.81 |
| Our model | **56.39** | **70.43** | **79.19** |

Figure 2: Word-level translation results, in precision at $k$. Results are averaged over 30 translation cases (15 two-way pairs).

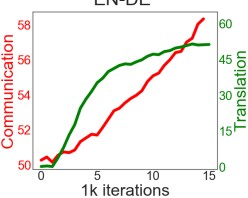

Figure 3: Learning curve for the EN-DE word-level model.

**Qualitative analysis**   As our agents learn foreign words by grounding them in visual space, we expect the learned foreign word embeddings to be semantically similar to corresponding images. We inspect the nearest neighbors of foreign word embeddings in each language, and find that concepts

---

[2]We experimented with more distractors, but found this setting to be optimal. See Appendix D for a discussion.

with similar images indeed have close word embeddings. See Appendix B for a discussion and relevant examples.

## 5 SENTENCE-LEVEL EXPERIMENTS

**Task**    We next train our models on a sentence-level communication task where agent $P_A$ is given an image, and needs to communicate its content in a sentence in its language $L_A$ to allow agent $P_B$ to identify the right image from a set of distractors (see §3.1).

**Datasets and preprocessing**    We use three datasets of images with annotations in multiple languages. The Multi30k (Elliott et al., 2016) dataset contains 30k images and two types of bilingual annotations for two different tasks:

- (Task 1) English-German translation task, where this can be aided by images; and
- (Task 2) German image captioning task, where this can be helped with English captions.

Training data for Task 1 consists of 1 English caption and its German translation for every image, translated by a professional translator. For Task 2, five English and five German captions are collected independently for every image. We use the original data split: 29k training, 1k validation and 1k test images.

We experiment with another language pair: English-Japanese. We use MS COCO (Lin et al., 2014; Chen et al., 2015), which contains 120k images and 5 English captions per image, and STAIR (Yoshikawa et al., 2017), a collection of Japanese annotations of the same dataset (also 5 per image). Following Karpathy & Li (2015), we use 110k training, 5k validation and 5k test images.

To ensure no parallel corpus is used to train our models, we partition the images in the training set into two parts (one for each langauge) and only use captions in one language for each half and not the other. With Multi30k, for instance, we have 14.5k English training images (whose German captions we discard) and 14.5k German training images.

We use tokenized Japanese captions in STAIR.[3] We lowercase, normalize and tokenize English and German captions using preprocessing scripts from Moses.[4] In addition, we tokenize German captions into subword symbols using the byte pair encoding (BPE) algorithm with 10k merge operations (Sennrich et al., 2015).

**Baselines**    We compare against several baselines that similarly only make use of disjoint image-description data. In increasing order of sophistication:

**Nearest neighbor**    To translate an English sentence into German, we use its corresponding image to find the closest image in our German training set. We then retrieve all corresponding German captions and compute BLEU score against the ground truth German test captions. This model is similar to our word-level nearest neighbor baselines.

**NMT with neighboring pairs**    Given our non-aligned training set of English and German image captions, without any overlapping images, we can form new EN-DE sentence pairs by finding the closest German training image for every English training image. We then pair every corresponding German caption with every corresponding English caption, and train a standard NMT model without attention (Cho et al., 2014; Sutskever et al., 2014) on these pairs. We do not compare against an NMT model with attention because our models do not use attention (since incorporating attention would mean that agents have access to each other's hidden states, which is no longer a multi-agent setting).

**N&N**    We implement and train end-to-end models from (Nakayama & Nishida, 2017). Their **two-way** model learns separate encoders to align the source language and images in a multimodal space. Then, a captioning model in the target language is trained on image representations, and is used to decode source representations to translate them. Their **three-way** models

---

[3]https://github.com/STAIR-Lab-CIT/STAIR-captions
[4]https://github.com/moses-smt/mosesdecoder

align both source and *target* languages with images using a target language encoder. Their models are similar to our models, with two key differences: (1) they are trained on a fixed corpus, without interaction between agents or learned communication, and (2) their model unit-normalizes the output of every encoder and is trained on pairwise ranking loss. In order to specifically examine the effectiveness of communication in learning to translate, we train these baselines using both their original loss function and our own loss function (see Appendix A).

**Models** In our base model, the agents learn to speak their native languages simultaneously as they learn to communicate with each other **(not pretrained)**. In a sense, this can be seen as a *tabula rasa* situation where both agents start from a blank slate. However, we also experiment with agents who *already speak* their languages, by using the weights from pretrained image captioning models in both languages to initialize our speaker modules and image encoders. Furthermore, we can freeze the parameters of image encoders or speaker modules to investigate their impact on communication and translation performance. In the most extreme case, where we pretrain and fix the speaker modules and image encoders **(pretrained, spk & enc fixed)**, we only train the foreign language encoder, using only the listener loss. All other models are trained on both the speaker and listener loss.

**Experimental settings** We train with 1 distractor ($K = 2$)[5] and minibatch size 64. The hidden state size and embedding dimensionalities are 1024 and 512, respectively. The learning rate and dropout rate are tuned on the validation set for each task. The vocabulary sizes of each language used in our experiments are: EN (4k) and DE (5K) for Multi30k Task 1, EN (8k) and DE (13k) for Multi30k Task2 and EN (10k) and JP (13k) for MS COCO.

We train the model on the communication task, and early stop when the validation translation BLEU score stops improving. We use beam search at inference time, with beam width tuned on the validation set.

| | | | Multi30k Task 1 | | Multi30k Task 2 | | COCO & STAIR | |
| | | | EN-DE | DE-EN | EN-DE | DE-EN | EN-JA | JA-EN |
|---|---|---|---|---|---|---|---|---|
| | | Nearest neighbor | 1.41 | 1.77 | 3.75 | 5.87 | 15.88 | 10.94 |
| | | NMT with neighboring pairs | 3.07 | 3.41 | 6.83 | 14.78 | 32.17 | 22.39 |
| Unaligned Models | Baselines — Their loss | N&N, 2-way, img | 2.57 | 2.69 | 5.22 | 12.78 | 28.68 | 20.61 |
| | | N&N, 3-way, img | 2.01 | 3.51 | 6.19 | 14.60 | 29.81 | 21.25 |
| | | N&N, 3-way, desc | 3.34 | 3.87 | 9.66 | 15.96 | 27.53 | 17.51 |
| | | N&N, 3-way, both | 1.50 | 3.62 | 9.89 | 15.50 | 31.01 | 20.59 |
| | Our loss | N&N, 2-way, img | 4.20 | 6.04 | 11.95 | 17.22 | 33.10 | 23.43 |
| | | N&N, 3-way, img | 2.32 | 5.91 | 11.62 | 17.84 | 32.11 | 23.61 |
| | | N&N, 3-way, desc | 5.13 | 6.02 | 11.07 | 17.01 | 26.65 | 17.82 |
| | | N&N, 3-way, both | 4.89 | 6.59 | 13.53 | 18.48 | 32.84 | 23.28 |
| | Our models | not pretrained | 5.80 | 7.20 | 14.81 | 17.70 | 33.26 | 23.66 |
| | | pretrained, spk & enc fixed | 5.81 | 7.36 | 13.87 | 18.68 | **35.25** | **24.61** |
| | | pretrained, spk fixed | **6.49** | **7.42** | **14.93** | **19.81** | 33.01 | 23.59 |
| | | pretrained, not fixed | 5.02 | 6.06 | 13.44 | 17.41 | 33.58 | 23.19 |
| Aligned NMT | | | 17.21 | 16.65 | 19.99 | 21.44 | 38.55 | 28.36 |

Table 1: Test BLEU scores for each model and dataset. The best performing (unaligned) model for each dataset is shown in bold. We show the results of baselines from (Nakayama & Nishida, 2017) using two different loss functions (Appendix A). *Pretrained* denotes initializing the speaker modules and image encoders with pretrained image captioning models. *Fixed* denotes fixing the parameters of either the speaker module or the image encoder.

**Results** We find that naïvely looking up the nearest training image and retrieving its captions gives relatively poor BLEU scores (Table 1, Nearest neighbor). On the other hand, training an NMT model on these visually closest neighbor pairs gives much better translation performance.

---

[5]See Appendix D for a discussion on the number of distractors.

From the results of baselines from (Nakayama & Nishida, 2017), it is clear that our loss function gives better performance than the pairwise ranking loss with unit-normalized encoder outputs. We note that these baselines perform worse than our models even when our loss function is used, an indication that communication helps in learning to translate. We conjecture that our listeners become better at aligning multimodal representations compared to these baselines, as our listeners are trained on speaker's output, and hence are exposed to a bigger and more diverse set of image descriptions. In contrast, the N&N models only make use of the ground truth captions. We also note that their 3-way models have an additional encoder for the target language, which our models lack. Although this is not used at test time, their 3-way models have 33% more parameters to train (97m) than our models (73m).

In contrast to (Nakayama & Nishida, 2017), where the best performance was obtained with end-to-end trained models, we find that our models benefit from initializing weights with pretrained captioning models. The model with fixed speaker modules and non-fixed image encoders gave best results in two out of three datasets, even outperforming the (spk & enc fixed) model, which only produces messages that are trained to predict ground truth captions. This shows that learning to send messages differently from the pretrained image captioning models achieves better translation performance than learning to send ground truth captions. We provide sample translations for our baselines and models in Appendix C. We also qualitatively examine completely zero-resource German-Japanese translation in Appendix F.

We compare our results with a standard non-attentional NMT model trained on parallel data (Cho et al., 2014; Sutskever et al., 2014). On COCO & STAIR, we observe a gap of approximately 4 BLEU compared to our best models On Multi30k Task 2, which is slightly smaller, the gap grows up to 5 BLEU scores. On Multi30k Task 1, where the dataset is the smallest and also of the highest quality (annotated by professional translators), the gap is around 11 BLEU. In other words, we find that our approach performs closer to supervised NMT as more training data is available, but that there still is a gap, which is unsurprising given the lack of parallel data.

**Qualitative analysis** We conjecture that our models learn to translate by having a shared visual space to ground source and target languages onto. Indeed, we show that our translation system fails without a common visual modality (see Appendix D). We note that using a larger number of distractors helps the model learn faster initially, but does not affect translation performance (see also Appendix D).

As expected, our models struggle with translating abstract sentences, although we observe that they can capture some visual elements in the source sentence (see Appendix E). This observation applies to most current grounded NMT systems, and it is an avenue worth exploring in future work but beyond the scope of the current work.

Inspired by the movie *Arrival* (2016), we show that our agents can learn to play the referential game, and learn to translate, using an alien language (Klingon) with only a small number of captions (see Appendix G). This example is meant to illustrate the point that our models can learn to translate even in situations where there is no knowledge whatsoever of the other language, and where training a professional translator would potentially take a long time.

## 6 MULTILINGUAL COMMUNITY OF AGENTS

**Task and dataset** Humans learn to speak languages within communities. We next investigate whether we can learn to translate better in a community of different language speakers, where every agent interacts with every other agent. We use the recently released multilingual Multi30k Task 1, which contains annotations in English, German and French for 30k images (Elliott et al., 2017). We train a community of three agents (each speaking one language) and let each agent learn the other two languages simultaneously.

We again partition the set of images into two halves (M1 and M2 in Figure 4), and ensure that the speaker and the listener do not see the same set of images (see Figure 4a for example). We experiment with two different settings: 1) a full community model, where having a multilingual community allows us to expose agents to more data than in the single-pair model while maintaining disjoint sets of images between agents (Figure 4c); and 2) a fair community model where the agents

are trained on exactly the same number of training examples (Figure 4b). We point out that the difference in training data should mainly affect the speaker module; the image encoder and foreign language encoder are trained on the same number of examples in all the models.

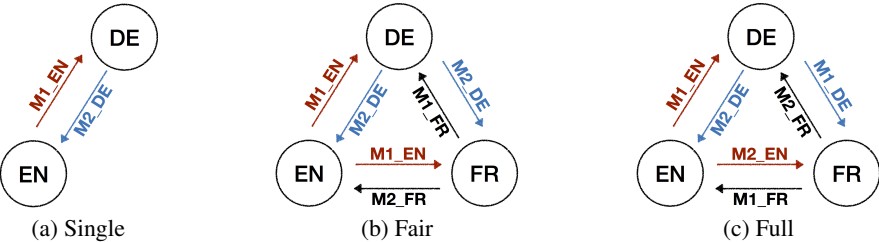

| (a) Single | (b) Fair | (c) Full |

Figure 4: Training data in single-pair and community models. **M1_EN** denotes the English annotations for the first half images in Multi30k. Red and blue indicate training data for the English and the German agents' speaker modules, respectively. Note that compared to the single pair model, English and German speakers see twice the amount of training data in the full model, but see the same number of examples in the fair model.

**Experimental settings**  We train our base model (not pretrained) on a three-way sentence-level communication task. We sample a language pair with an equal probability every minibatch and let the two agents communicate. Every agent has one image encoder, one native speaker module and two foreign language encoders. We tokenize every corpus using BPE with 10k merge operations. We use the same architecture for the three models: $D_{emb} = 128, D_{hid} = 256$ and we use a learning rate of 3e-4 and batch size of 128.

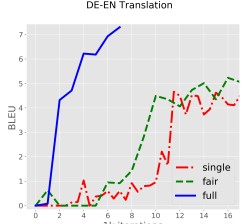

Figure 5: DE-EN learning curve for different models.

| Model | EN-DE | DE-EN | EN-FR | FR-EN | DE-FR | FR-DE |
|-------|-------|-------|-------|-------|-------|-------|
| Single | 3.85 | 5.36 | 5.20 | 5.87 | 4.31 | 3.92 |
| Fair | 3.73 | 5.56 | 4.81 | 5.96 | 5.08 | 4.00 |
| Full | **4.83** | **7.21** | **7.09** | **8.10** | **6.55** | **5.15** |

Table 2: Multi30k Task 1 Test BLEU scores. Results should be compared with the first two columns in Table 1.

**Results**  We observe that multilingual communities learn better translations. By having access to more target-side data, the full community model achieves the best translation performance in every language pair (Table 2). The fair community model achieves comparable performance to the single pair model.

We show the learning curves of different models in Figure 5. The full community model clearly learns much faster than the other two. The fair community also learns faster than the single-pair model, as it learns with equivalent speed but with less exposure to individual language pairs, since we sample a language pair for each batch, rather than always having the same one.

## 7 CONCLUSIONS AND FUTURE WORK

In this paper, we have shown that the ability to understand a foreign language, and to translate it in the agent's native language, can emerge from a communication task. We argue that this setting is natural, since humans learn language in a similar way: by trying to understand other humans while being grounded in a shared environment.

We empirically confirm that the capability to translate is facilitated by the fact that agents have a shared visual modality they can refer to in their respective languages. Our experiments show that our model outperforms recently proposed baselines where agents do not communicate, as well as several nearest neighbor based baselines, in both sentence- and word-level scenarios.

In future work, we plan to examine how we can enrich our agents with the ability to understand and translate abstract language, possibly through multi-task learning.

ACKNOWLEDGEMENT

KC thanks support by eBay, TenCent, Facebook, Google and NVIDIA. We thank our colleagues from FAIR for helpful discussions.

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

## A    CHOICE OF LISTENER LOSS FUNCTION

We compare two alternatives for training the listener: (1) a pairwise ranking loss, as used in Havrylov & Titov (2017) and Nakayama & Nishida (2017), and (2) the MSE loss, in addition to the one (Eq. (1)) used in this paper.

**Pairwise ranking loss**    Denoting the message vector, the target image and the $k$-th distractor image as $\hat{m}, i$ and $i_k$, respectively, this cost function is expressed as:

$$\mathcal{J}_{(\text{lsn, rank})} = \sum_{k=1}^{K} \max\left(0, \alpha - \text{sim}\big(E_{\text{EN}}^B(\hat{m}), E_{\text{IMG}}^B(i)\big) + \text{sim}\big(E_{\text{EN}}^B(\hat{m}), E_{\text{IMG}}^B(i_k)\big)\right),$$

where $\alpha$ is a margin hyperparameter and $\text{sim}(\cdot)$ is a function that computes the similarity between two vectors. The most common choice for $\text{sim}(\cdot)$ in practice is cosine similarity (Nakayama & Nishida, 2017). Note, however, that this only aligns the direction of the message and the target image vectors, not the magnitude. To facilitate translation, the speaker module should take as input a normalized image vector. We found this use of normalized image vectors to consistently hurt performance in all our experiments.

**MSE loss**    It has been found that minimizing the MSE loss can be effective in learning visually grounded representations of language (see, e.g., Chrupala et al., 2015). While our loss function is very similar to minimizing the MSE loss, there is an important distinction. Letting $x = E_{\text{EN}}^B(\hat{m}) - E_{\text{IMG}}^B(i)$, the MSE loss function is given by $x^2$, with a derivative of $2x$. Our loss function is $-\log(1/x^2)$, of which the derivative is $2/x$. Note that the gradient is initially small, when $x$ is large, but grows larger in magnitude as the listener loss is minimized.

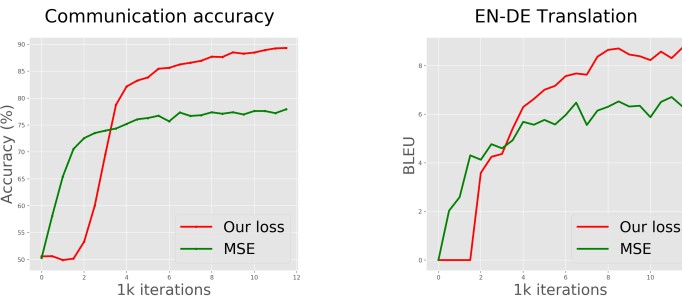

Figure 6: Comparison of our listener loss function with MSE loss.

In Figure 6, we show the learning curve of our model on Multi30k Task 2 EN-DE trained using either our listener loss function or the MSE loss. At the initial stage of learning, the listener trained using our loss function learns very slowly due to a small gradient. As the listener becomes better at aligning target images and sentences, it begins to learn much more quickly. On the other hand, we observe that the listener trained using the MSE loss immediately starts to learn, but learning slows down quickly.

Our loss term $-\log(1/x^2)$ is not defined at 0. In practice, $x$ is almost never 0, and we further add small noise to $x$ both to avoid $x = 0$ and to regularize learning. Empirically, we observe our formulation gives much better translation performance than both the pairwise ranking loss and the MSE loss.

## B    WORD-LEVEL NEAREST NEIGHBOR ANALYSIS

Table 3 showcases three concepts in English and German, where for each concept, the most representative image is shown, as well as the five nearest neighboring words (by cosine similarity).

We note that nearest neighbors for most words correspond to our semantic judgments: *galaxy* is closest to *universe* in English and *Galaxie* to *Universum* in German. Similarly, *plant* or *Pflanze*

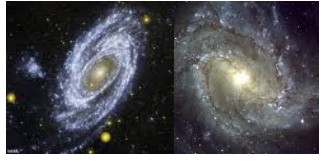  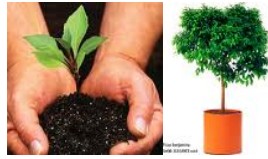  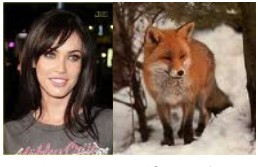

(a) Galaxy        (b) Galaxie                (c) Plant    (d) Pflanze              (e) Fox      (f) Fuchs

| | Word | Neighbors |
|---|---|---|
| (a) | Galaxy | Universe, Comet, Meteor, Exoplanet, Planet |
| (b) | Galaxie | Universum, Exoplanet, Komet, Meteor, Planeten |
| (c) | Plant | Leaf, Flower, Sunflower, Tree, Cucumber |
| (d) | Pflanze | Blatt, Gurke, Blume, Baum, Garten |
| (e) | Fox | Celebrity, Girl, Hair, Woman, Bell |
| (f) | Fuchs | Känguru, Löwe, Hirsch, Esel, Wolf |

Table 3: Nearest neighbors of foreign word embeddings learned from communication, along with a sample image for each concept in the dataset. The English word embeddings were learned by the German agent and vice versa.

is closest to *leaf* or *Blatt*. For concepts that evoke different senses across languages, however, we observe different nearest neighbors. For example, most images for *Fox* in the dataset contain a person, whereas images for *Fuchs* contain an animal. This encourages the German agent to associate *Fox* closely with *Celebrity* and *Girl*, whereas the English agent learns that *Fuchs* is a furry animal, similar to *Känguru* or *Löwe*.

## C    SENTENCE-LEVEL SAMPLE TRANSLATIONS

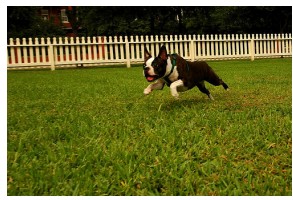  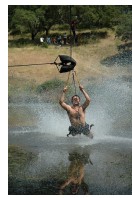  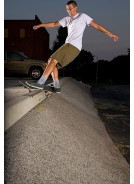

| Src | ein hund springt auf einer wiese vor einem weißen zaun in die luft . | ein mann hängt an einem seil mit rollen das über ein wasser gespannt ist . | ein skateboarder an einer böschung zu einem parkplatz . |
|---|---|---|---|
| Ref | a dog runs on the green grass near a wooden fence . | a man wearing bathing trunks is parasailing in the water . | a skateboard is grinding on a curb with his skateboard . |
| NN | a brunette photographer is kneeling down to take a photo . | police watch some punk rock types at a protest . | a man is standing on the streets taking photographs . |
| NMT | a dog is jumping over a fence . | a man in a blue shirt is riding a bike . | a man in a blue shirt is riding a bike . |
| N&N | a brown dog is running on the grass . | a man in a wetsuit is surfing a large wave . | a man in a blue shirt is walking down the sidewalk . |
| Model | two dogs playing with a ball in the grass . | a man is parasailing in the ocean . | a man in a blue shirt and black pants is skateboarding . |

Table 4: Sample DE-EN translations from Multi30k Task 2 test set. Images were not used to aid translation and are only shown for references. We show the source sentence as *Src* and one of the five target sentences as *Ref*. The outputs from the nearest neighbor baseline are shown as *NN*, NMT baseline with neighbor pairs as *NMT*, the 3-way, both decoder model from (Nakayama & Nishida, 2017) as *N&N*, and our (pretrained, spk fixed) model as *Model*.

In Table 4, we compare sample translations from our nearest neighbor and NMT baselines, as well as the best model from (Nakayama & Nishida, 2017) and our best model. We observe that the nearest neighbor baseline generates translations that are mostly unrelated to the reference sentence. The NMT baseline, on the other hand, often generates the correct subject of the sentence, and seems capable of capturing the main actor in the scene. The 3-way, both decoder model from (Nakayama & Nishida, 2017) captures the main actors and the environment in the scene. Our model appears to

capture even the minor details, such as quantity (number of dogs) and specific activity (parasailing, skateboarding).

## D    SENTENCE-LEVEL QUALITATIVE ANALYSIS

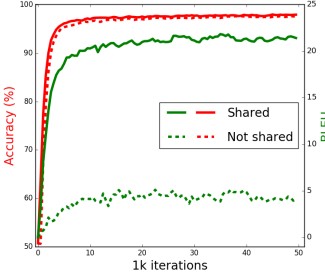

Figure 7: Sharing $E^I$

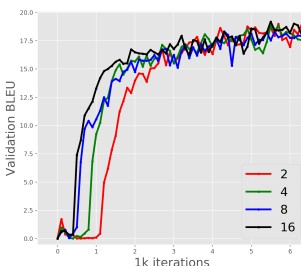

Figure 8: Number of distractors.

We conjectured that grounding the native and foreign languages into a shared visual space allows agents to understand foreign languages. To verify this, we trained our base model (not pretrained) on COCO & STAIR, where agents have language-specific image encoders, e.g. the English agent has two separate image encoders: one for the English speaking module and another for the Japanese encoder. We compare this with our original model with the same architecture. We plot the communication accuracy and JA-EN validation BLEU scores in Figure 7. Red curves indicate validation communication accuracy (left axis), and green curves indicate BLEU score (right axis). Solid lines denote the standard model and dotted lines denote the model without sharing the image encoder.

We observe no significant difference in the communication performance: the model that does not share the image encoder performs just as well in the communication task. However, translation performance greatly suffers without access to the shared visual modality.

In Figure 8, we show the learning curve of four base models, with different number of distractors $(K)$. We observe that a larger number of distractors helps the model learn faster initially, but otherwise gives no performance benefit.

## E    TRANSLATION IN NON-CONCRETE DOMAINS

As our agents understand foreign languages through grounding in the visual modality, we investigate their ability to generalize to non-visual, abstract domains. We train our (pretrained, spk fixed) model on Multi30k Task 2, and let it translate German sentences from WMT'15 DE-EN validation and test set. See Table 5.

We note that our model is able to capture some visual elements in a sentence, such as *snow* or *mountain*, but generally produces poor quality translations. We observe that most words in the source sentence from WMT'15 do not occur in Multi30k's training set, hence our model mostly receives $<UNK>$ vectors.

## F    ZERO-RESOURCE TRANSLATION

To showcase our models' ability to learn to translate without parallel corpora, we train our base model on a communication task between a low-resource language pair: German and Japanese. We take the German corpus from Multi30k Task2, the Japanese corpus from STAIR, and train two models on a sentence-level communication task between the two languages. In Tables 6 and 7, we show the Japanese source sentence (src), the model output in German (hyp), and their translation to English using Google Translate. We observe that our model mostly generates reasonable sentences, and captures properties such as color and action in the scene.

| Src | Schnee liegt insbesondere auf den Straßen im Riesengebirge und im Gebirge Orlické hory. |
|-----|------------------------------------------------------------------------------------------|
| Ref | Snow is particularly affecting the roads in the Krkonose and Orlicke mountains. |
| Hyp | a man is standing on a snow covered mountain . |

| Src | Das Tote Meer ist sogar noch wärmer und dort wird das ganze Jahr über gebadet. |
|-----|------------------------------------------------------------------------------------------|
| Ref | The Dead Sea is even warmer, and people swim in it all year round. |
| Hyp | a man is surfing in the ocean . |

| Src | Es folgten die ersten Radfahrer und Läufer um 10 Uhr. |
|-----|------------------------------------------------------------------------------------------|
| Ref | Then it was the turn of the cyclists and runners, who began at 10 am. |
| Hyp | a man in a red and white uniform is riding a dirt bike in front of a crowd . |

| Src | Das Baby, das so ergreifend von einem Spezialeinsatzkommando in Sicherheit getragen wurde |
|-----|------------------------------------------------------------------------------------------|
| Ref | The baby who was carried poignantly to safety by a special forces commando |
| Hyp | a baby in a yellow shirt is sleeping . |

Table 5: Sample translations from WMT'15 DE-EN validation and test set.

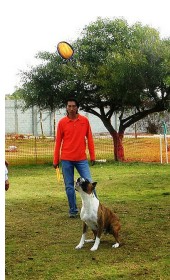 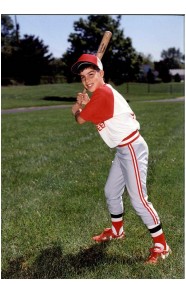 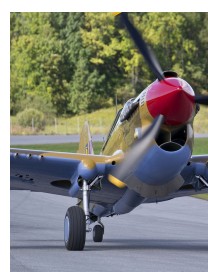

| Src | フリスビー を キャッチ しよう と 構える 犬 | 赤 と 白 の 野球 の ユニフォーム を 着た 少年 が バット を 構えて いる | 黄色い プロペラ 機 が 離陸 する 瞬間 |
|-----|------|------|------|
| Hyp | ein hund fängt einen frisbee . | ein baseballspieler mit rotem helm und weißem trikot holt mit dem schläger aus , um den heranfliegenden ball zu treffen . | ein flugzeug fliegt über eine flugzeug-piste . |
| Src (en) | Dog keeps trying to catch Frisbee | A boy in a red and white baseball uniform wears a bat | The moment when a yellow propeller plane takes off |
| Hyp (en) | a dog catches a frisbee. | a baseball player with a red helmet and a white tricot takes the bat out to hit the ball. | An airplane flies over an airplane runway. |

Table 6: Sample JA-DE translations

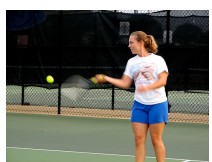 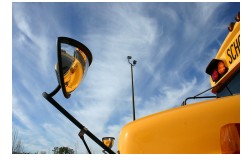 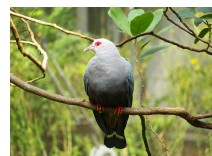

| Src | 黄色い テニス ボール を 打ち返す 女性 | 黄色い バス の フロント 部分 に 取り付け られた バックミラー に 黄色い バス が 映って いる | 木 の 枝 に 止まって いる 目 の 周り と 足 が 赤い 鳥 |
|-----|------|------|------|
| Hyp | eine tennisspielerin in weiß und grünem oberteil spielt tennis | ein gelber bus steht geparkt vor einem nicht fertigen gebäude . | ein vogel sitzt auf einem ast . |
| Src (en) | A woman striking a yellow tennis ball | A yellow bus is reflected on the rearview mirror attached to the front part of the yellow bus. | Around the eyes stopping at the branches of the tree and a red bird |
| Hyp (en) | a tennis player in white and green shell playing tennis | A yellow bus is parked in front of a non-finished building. | A bird sitting on a branch. |

Table 7: Sample JA-DE translations

## G   ALIEN LANGUAGE TRANSLATION

To demonstrate our models' ability to learn to translate only with monolingual captions, we experiment with a language for which no parallel corpus exists, nor the knowledge of the language itself: Klingon. As no image captions are available in Klingon, we translate 15k English captions in Multi30k Task 1 into Klingon pIqaD[6] using Bing Translator.[7] We tokenize the Klingon captions and discard words occurring less than 5 times in the training data. We then train our base model (no pretraining) on English and Klingon communication. In Tables 9 and 10, the source sentence in English is shown as *src*, the Klingon model output in *hyp*, and the English translation of the output in *hyp (en)* (using Bing Translator). Although the Klingon training data is noisy and imperfect, we observe that our model learns to translate only with 15k Klingon captions. This example illustrates how we can learn to translate even if there is no knowledge of the other language, and where a professional translator would take a long time to first acquire the other language.

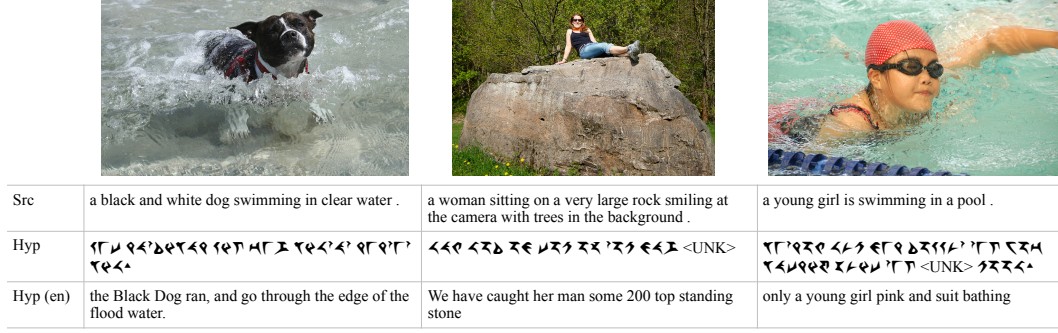

| Src | a black and white dog swimming in clear water . | a woman sitting on a very large rock smiling at the camera with trees in the background . | a young girl is swimming in a pool . |
|-----|---|---|---|
| Hyp | ꝪꝹ ꝺꝺꝺꝺꝺꝺ ꝺꝺꝹ ꝺꝺ ꝺ ꝺꝺꝺꝺ ꝺꝺꝺꝺ ꝺꝺꝺꝺ | ꝺꝺꝺ ꝺꝺꝺ ꝺꝺ ꝺꝺꝺ ꝺꝺ ꝺꝺ ꝺꝺꝺꝺ <UNK> | ꝺꝺꝺꝺꝺ ꝺꝺꝺ ꝺꝺꝺ ꝺꝺꝺꝺꝺ ꝺꝺ ꝺꝺ ꝺꝺꝺ ꝺꝺꝺꝺꝺꝺ ꝺꝺꝺꝺ ꝺꝺ ꝺꝺ <UNK> ꝺꝺꝺꝺ |
| Hyp (en) | the Black Dog ran, and go through the edge of the flood water. | We have caught her man some 200 top standing stone | only a young girl pink and suit bathing |

Figure 9: Sample English-Klingon translations

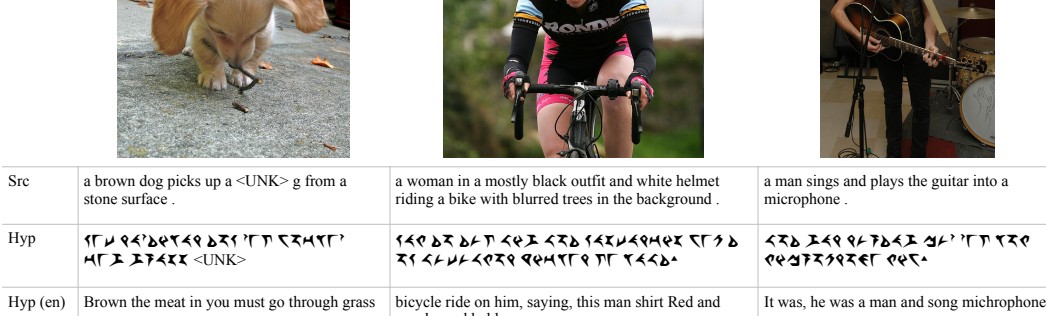

| Src | a brown dog picks up a <UNK> g from a stone surface . | a woman in a mostly black outfit and white helmet riding a bike with blurred trees in the background . | a man sings and plays the guitar into a microphone . |
|-----|---|---|---|
| Hyp | ꝪꝹ ꝺꝺꝺꝺꝺꝺ ꝺꝺꝺ ꝺꝺ ꝺꝺꝺꝺꝺꝺ ꝺꝺ ꝺ ꝺꝺꝺꝺꝺ <UNK> | ꝺꝺꝺ ꝺꝺ ꝺꝺ ꝺꝺꝺ ꝺꝺꝺ ꝺꝺꝺꝺꝺꝺꝺꝺ ꝺꝺꝺ ꝺꝺ ꝺꝺꝺꝺꝺꝺꝺ ꝺꝺꝺꝺꝺ ꝺ ꝺꝺ ꝺꝺꝺꝺ | ꝺꝺꝺ ꝺꝺꝺ ꝺꝺꝺꝺꝺꝺ ꝺꝺ ꝺꝺ ꝺꝺꝺ ꝺꝺꝺꝺꝺꝺꝺꝺꝺꝺ ꝺꝺꝺꝺ |
| Hyp (en) | Brown the meat in you must go through grass | bicycle ride on him, saying, this man shirt Red and purple, and bald. | It was, he was a man and song microphone. |

Figure 10: Sample English-Klingon translations

---

[6]https://en.wikipedia.org/wiki/Klingon_alphabets
[7]https://www.bing.com/translator

