# OpenReview forum: "Emergent Translation in Multi-Agent Communication"
_ICLR.cc/2018/Conference — Accept (Poster)_

### Official Review · AnonReviewer1 · 2017-11-27
**Good paper. Please change the title.**

**Rating:** 8
**Confidence:** 5

**Review:**

Summary:

This paper proposes a multi-agent communication task where the agents learn to translate as a side-product to solving the communication task. Authors use the image modality as a bridge between two different languages and the agents learn to ground different languages to same image based on the similarity. This is achieved by learning to play the game in both directions. Authors show results in a word-level translation task and also a sentence-level translation task. They also show that having more languages help the agent to learn better.

My comments:

The paper is well-written and I really enjoyed reading this paper. While the idea of pivot based common representation learning for language pairs with no parallel data is not new, adding the communication aspect as an additional supervision is novel. However I would encourage authors to rephrase their claim of emergent translation (the title is misleading) as the authors pose this as a supervised problem and the setting has enough constraints to learn a common representation for both languages (bridged by the image) and hence there is no autonomous emergence of translation out of need. I see this work as adding communication to improve the translation learning.

Is your equation 1 correct? I understand that your logits are reciprocal of mean squared error. But don’t you need a softmax before applying the NLL loss mentioned in equation 1? In current form of equation 1, I think you are not including the distractor images into account while computing the loss? Please clarify.

What is the size of the vocabulary used in all the experiments? Because Gumbel Softmax doesn’t scale well to larger vocabulary sizes and it would be worth mentioning the size of your vocabulary in all the experiments.

Are you willing to release the code for reproducing the results?

Minor comments:

In appendix C, Table 4 caption: you say target sentence is “Trg” but it is “Ref” in the table. Also is the reference sentence for skateboard example typo-free?

---

> ### Author Response · Authors · 2017-12-26
> **Thank you for the helpful comments!**
>
> - Emergent translation?
>
> By “emergent translation”, we meant that translation emerges as a consequence of having two agents solve a referential game, without parallel corpora. The referential game involves images and languages, but the translation emerges between language and language - an emergent property of the combination of the objective function and the model weight tying (specifically that both languages used by the speaker use the same visual system/weights).
>
> Is this explanation satisfactory? Otherwise, do you have any suggestions?
>
> - Equation 1
>
> Thanks for spotting this. Yes softmax was indeed used, so the distractor examples were penalized via partition function of the softmax. This is fixed in the revision.
>
> - Vocabulary sizes
>
> Multi30K Task 1 : (EN: 4035, DE : 5445)
> Multi30K Task 2 : (EN: 8618, DE : 13091)
> COCO : (JP : 13019, EN : 10396)
>
> We added this in the revision.
>
> - Open-sourcing the code
>
> Yes, we plan to open-source the code shortly.
>
> - Typo in Table 4
>
> Thanks for spotting the typo, yes the reference is typo free but we accidentally put in a wrong image. This is fixed it in our revision.

---

> > ### Comment · AnonReviewer1 · 2018-01-12
> > **still not convinced with the title**
> >
> > My reply to authors' arguments about emergent translation:
> >
> > I agree that the agents learn to translate without seeing any parallel data. But you are bridging the languages through image which is the common modality. How is this different than bridge based representation learning or machine translation? The only novelty here is you add communication as an extra supervision to the bridge based MT. I am still against the usage of the word "emergent". You can motivate this work as communication for extra supervision in a bridge based MT.
> >
> > Nevertheless, I have given 8/10 for this paper since this deserves to be accepted. I am happy with other responses for my review.

---

### Official Review · AnonReviewer3 · 2017-11-27

**Rating:** 7
**Confidence:** 3

**Review:**

--------------
Summary and Evaluation:
--------------
This work present a novel multi-agent reference game designed to train monolingual agents to perform translation between their respective languages -- all without parallel corpora. The proposed approach closely mirrors that of Nakayama and Nishida, 2017 in that image-aligned text is encouraged to map to similarly to the grounded image. Unlike in this previous work, the approach proposed here induces this behavior though a multi-agent reference game. The key distinction being that in this gamified setting, the agents sample many more descriptions from their stochastic policies than would otherwise be covered by the human ground truth. The authors demonstrate that this change results in significantly improved BLEU scores across a number of translation tasks. Furthermore, increasing the number of agents/languages in this setting seems to

Overall I think this is an interesting paper. The technical novelty is somewhat limited to a minor (but powerful) change in approach from Nakayama and Nishida, 2017; however, the resulting translators outperform this previous method. I have a few things listed in the weaknesses section that I found unclear or think would make for a stronger submission.


--------------
Strengths:
--------------

- The paper is fairly clearly written and the figures appropriately support the text.

- Learning translation without parallel corpora is a useful task and leveraging a pragmatic reference game to induce additional semantically valid samples of a source language is an interesting approach to do so.

- I'm also excited by the result that multi-agent populations tend to improve the rate of convergence and final translation abilities of these models; though I'm slightly confused about some of the results here (see weaknesses).

--------------
Weaknesses:
--------------

- Perhaps I'm missing something, but shouldn't the Single EN-DE/DE-EN results in Table 2 match the not pretrained EN-DE/DE-EN Multi30k Task 1 results? I understand that this is perhaps on a different data split into M1/2 but why is there such a drastic difference?

- I would have liked to see some context as how these results compare to an approach trained with aligned corpora. Perhaps a model trained on the human-translated pairs from Task 1 of Multi30k? Obviously, outperforming such a model is not necessary for this approach to be interesting, but it would provide useful context on how well this is doing.

- A great deal of the analysis and qualitative examples are pushed to the supplement which is a bit of a shame given they are quite interesting.

---

> ### Author Response · Authors · 2017-12-26
> **Thank you for the helpful comments!**
>
> - Performance difference in Table 1 and 2
>
> The size of the model used is different between Table 1 and Table 2. D_hid and D_emb are (1024, 512) for the models in Table 1, and (256, 128) for the models in Table 2. Also, the community models were early stopped based on the overall performance across 6 different language pairs, instead of two (as was the case for single models), which could have also caused the difference in BLEU score.
>
> - Comparison with models trained on fully parallel corpora
>
> We added the performance obtained by an NMT model trained on aligned corpora in Table 1.
>
> - Appendix
>
> We agree. Due to page constraints we were forced to move many interesting analyses to the appendix.

---

### Official Review · AnonReviewer2 · 2017-11-28
**Needs better writing, better comparisons**

**Rating:** 5
**Confidence:** 5

**Review:**

Summary: The authors show that using visual modality as a pivot they can train a model to translate from L1 to L2.

Please find my detailed comments/questions/suggestions below:

1) IMO, the paper could have been written much better. At the core, this is simply a model which uses images as a pivot for learning to translate between L1 and L2 by learning a common representation space for {L1, image} or {L2, image}. There are several works on such multimodal representation learning but the authors present their work in a way which makes it look very different from these works. IMO, this leads to unnecessary confusion and does more harm than good. For example, the abstract gives an impression that the authors have designed a game to collect data (and it took me a while to set this confusion aside).

2) Continuing on the above point, this is essentially about learning a common multimodal representation and then decode from this common representation. However, the authors do not cite enough work on such multimodal representation learning (for example, look at Spandana et. al.: Image Pivoting for Learning Multilingual Multimodal Representations, EMNLP 2017 for a good set of references)

3) This omission of related work also weakens the experimental section. At least for the word translation task many of these common representation learning frameworks could have been easily evaluated. For example, find the nearest german neighbour of the word "dog" in the common representation space. The authors instead compare with very simple baselines.

4) Even when comparing with simple baselines, the proposed model does not convincingly outperform them. In particular,  the P@5 and P@20 numbers are only slightly better.

5) Some of the choices made in the Experimental setup seem questionable to me:
   - Why  use a NMT model without attention? That is not standard and does not make sense to use when a better baseline model (with attention) is available ?
   - It is mentioned that "While their model unit-normalizes the output of every encoder, we found this to consistently hurt performance, so do not use normalization for fair comparison with our models." I don't think this is a fair comparison. The authors can mention their results without normalization if that works well for them but it is not fair to drop normalization from the model of N&N if that gives better performance. Please mention the numbers with unit normalization to give a better picture. It does not make sense to weaken an existing baseline and then compare with it.

6) It would be good to mention the results of the NMT model in Table 1 itself instead of mentioning them separately in a paragraph. This again leads to poor readability and it is hard to read and compare the corresponding numbers from Table 1.  I am not sure why this cannot be accommodated in the Table itself.

7) In Figure 2, what exactly do you mean by "Results are averaged over 30 translation scenarios". Can you please elaborate ?

---

> ### Author Response · Authors · 2017-12-26
> **Thank you for the helpful comments!**
>
> 1) Our approach differs from previous works on multimodal representation learning and translation in two ways:
>
> In the existing multimodal NMT setting, we are often given a set of images and their descriptions in both source and target languages, while our setting goes further by giving disjoint sets of image-text pairs to the agents.
>
> A key difference between our work and many previous works in multimodal representation learning and translation (including Nakayama and Nishida, 2017) is that our agents learn to translate from communicating with each other. This allows our agents to learn from a far more diverse set of image descriptions than otherwise available as ground truth captions. We show that adding the communication element leads to significantly improved BLEU scores across several translation tasks.
>
> 2) Thanks for pointing out. We cited previous relevant work on multimodal/multilingual representation learning in our revision.
>
> 3) Regarding the comment “these common representation learning frameworks could have been easily evaluated”, did you mean something like the Semantic Textual Similarity task? Using your example of translating the word “dog” to German, our model actually finds the nearest German word neighbour in the joint space. Given the particular dataset that we used (Bergsma500), nearest neighbour methods based on similarity in the ConvNet feature space were actually the only reasonable (and fairly strong) baselines we could think of. Do you have any other suggestions?
>
> 4) Note that Bergsma500 is a very small dataset (500 categories X 20 images). Considering that we halve our dataset to train each agent, the training data is indeed extremely small, which could have caused limited performance improvement over our baselines. We have tried pre-training our models on ImageNet and fine-tuning it on Bergsma. This performed better than training on Bergsma from scratch, but we did not include this in our paper.
>
> 5) -Q : Why was attentional NMT not used?
>
> Our model does not use attention, so we decided not to use attention in our baseline for fairness. Incorporating attention into our model is not trivial, as attention has to be performed over the image vectors from the image encoder. We leave this as future work.
>
> -Q : Why was N&N baseline with normalization not compared with?
>
> We tested both versions of Nakayama’s model (with and without normalization). Not using normalization consistently outperformed using normalization. So we left out the numbers for the model with normalization to strengthen our baseline (not weaken it). Nevertheless, we will include the results for Nakayama’s with normalization in the revision.
>
> 6) We added the NMT results into Table 1.
>
> 7) We have 15 language pairs in Bergsma500, and we train our model to communicate and translate in both directions (e.g. EN->DE and DE->EN). We averaged the Precisions @ K (K=1, 5, 20) across all language pairs.

---

### Author Response · Authors · 2018-01-05
**Thanks to reviewers for the constructive feedback!**

We have uploaded a new revision. The revision addresses the reviewer’s comments, with the following changes in particular:

1) Added more references on multimodal / multilingual representation learning in Section 2.

2) We explain (Section 5, “NMT with neighboring pairs”) why we do not compare against an NMT model with attention. The reason is that incorporating attention would mean that agents have access to each other's hidden states, which is no longer a multi-agent setting and outside of the scope of our work.

3) We added an (even) stronger comparison against Nakayama and Nishida, also including their original loss function with normalization in Table 1.

4) We added NMT results into Table 1 instead of having them separately as a paragraph.

5) In addition, we fixed a bug in our beam search code and updated the BLEU scores in Table 1 and 2 accordingly. We achieve higher BLEU scores, and the results stay the same: our models consistently outperform all our baselines.

---

### Decision · Program_Chairs · 2018-01-29
**ICLR 2018 Conference Acceptance Decision**

**Decision:**

Accept (Poster)

**Comment:**

The paper considers learning an NMT systems while pivoting through images. The task is formulated as a referential game. From the modeling and set-up perspective it is similar to previous work in the area of emergent communication / referential games, e.g., Lazaridou et al (ICLR 17) and especially to Havrylov & Titov (NIPS 17), as similar techniques are used to handle the variable-length channel (RNN encoders / decoders + the ST Gumbel-Softmax estimator).  However, its multilingual version is interesting and the results are sufficiently convincing (e.g., comparison to Nakayama and Nishida, 17). The paper would more attractive for those interested in emergent communication than the NMT community, as the set-up (using pivoting through images) may be perceived somewhat exotic by the NMT community. Also, the model is not attention-based (unlike SoA in seq2seq / NMT), and it is not straightforward to incorporate attention (see R2 and author response).

+ an interesting framing of the weakly-supervised MT problem
+ well written
+ sufficiently convincing results
- the set-up and framework (e.g., non-attention based) is questionable from practical perspective